# THEORETICAL PRINCIPLES OF MULTI-AGENT REINFORCEMENT LEARNING FOR COALITIONAL BARGAINING GAMES

**Lucia Cipolina-Kun**
University of Bristol. UK
`lucia.kun@bristol.ac.uk`

**Ignacio Carlucho**
Heriot-Watt University

**Stephen Mak**
University of Cambridge, UK

**Kalesha Bullard**
Deepmind

**Vahid Yazdanpanah, Enrico Gerding, Sebastian Stein**
University of Southampton. UK

## ABSTRACT

The rising focus on employing multi-agent reinforcement learning (MARL) in coalitional bargaining games (CBG) has exposed a need for robust theoretical principles linking the two. To address this, we explore the relationship between CBG and MARL within the context of stochastic games, and show that under some assumptions, CBG are a subclass of sequential stochastic games. Out work is a step forward in the reproducibility and generalization of MARL results to CBG.

## 1 INTRODUCTION AND RELATED LITERATURE

Coalition formation is a strategy in which self-interested agents work together to achieve greater rewards than what they could attain alone. A coalitional bargaining game (CBG) is a type of cooperative game that captures the process of negotiation among players in a coalition. By following a bargaining protocol, players seek to form stable agreements on the coalition teammates and distribution of payoffs. The number of negotiation rounds makes the coalitional bargaining process lengthy and potentially inefficient, thus, prior work introduced multi-agent reinforcement learning (MARL) to speed up the bargaining rounds of a CBG by creating negotiating agents (Bachrach et al., 2020; Chen et al., 2022; Hughes et al., 2020; Taywade, 2021; Chalkiadakis et al., 2011; Mak et al., 2021). In principle, MARL is a computational framework used to approximate the solution of stochastic games (Shapley, 1953; Littman, 1994)[1] and while the *application* of MARL to CBG has been abundant, there is no literature providing *theoretical grounding* for the connection between CBG and stochastic games. This is not an easy task since the fundamental components of the stochastic game's tuple (i.e., the state's transition dynamics and the rewards), are not explicitly defined for a CBG. In traditional game theoretic works such as Rubinstein (1982); Morgenstern (1973); Okada (1996) CBG are defined simply as *extensive form games* of perfect information. This is, agents are assumed to be fully rational and can predict the future with certainty. As such, there is no apparent formal connection between CBG and stochastic games allowing a principled use of MARL. Therefore, the application of MARL to CBG is hampered by the absence of a theoretical framework linking stochastic games and CBG; resulting in limited *reproducibility* and *generalization* of the results (aspects crucial for AI research Hutson (2018); Haibe-Kains et al. (2020)). The aim of this paper is to answer: given that MARL provides an empirical framework for stochastic games, which theoretical principles make it suitable for CBG? Which extra assumptions are needed for the relationship to hold?

**Contributions**. First, we introduce a *novel characterization* of CBG as extensive-form games that can be defined via a tuple representation. Second, we outline the *assumptions* under which CBG can be modelled as turn-based stochastic games. Third, we connect MARL to CBG in a principled way.

---

[1]For a connection between stochastic games and MARL refer to Appendix A.1.3.

## 2 CBG, STOCHASTIC GAMES AND MARL - A GOLDEN BRAID

We aim for a theoretical framework justifying the application of MARL to CBG in a principled way. Since the connection between MARL and CBG is not immediate, we prove that under certain conditions, CBS can be modelled as stochastic games, as shown on Figure 1.

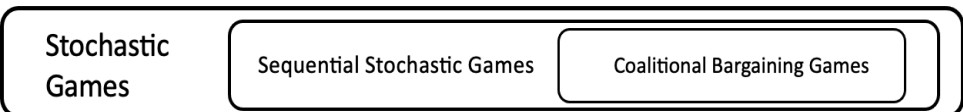

Figure 1: GBG can be modelled as sequential stochastic games under the assumption of stochastic transition dynamics and stochastic rewards..

In Definition 1 we propose a *novel* characterization of CBG as games that admit a tuple representation. The rationale is to extend this initial CBG definition to obtain first turn-based stochastic games (TBSG) and ultimately stochastic games as a superset of CBG.

**Definition 1 (Coalitional bargaining game)** *Consists of a tuple $(N, T, \mathcal{A}, \delta)$ where $N$ is the set of agents with $|N| = n$ the number of agents, $T$ is the maximum length of the game (can be infinite), $\mathcal{A} = \{\mathcal{A}_p \cup \mathcal{A}_r\}$ is the set composed of available actions in a bargaining round ($\mathcal{A}_p$ coalition proposals and $\mathcal{A}_r$ responses) and $\delta$ is the discount factor accounting for the value of time. The set of all bargaining games is denoted by $\mathcal{B}$. The solution is given by the function $f : \mathcal{B} \to 2^n$ that maps the set of proposals to the set of stable coalitions.*

The characterization of a CBG in Definition 1 follows the setting traditionally used by the game theory community (such as Rubinstein (1982); Okada (1996)) which involves no stochastic components or uncertainty. In this context, applying MARL to an already solved game [2] would seem unproductive and unprincipled as there is no *environment* (stochastic transition dynamics and rewards) involved. To progress towards the setting of stochastic games the following assumptions are needed: let $\mathcal{S}_t = \{\mathcal{S}_p \cup \mathcal{S}_r\}$ denote the states of a CBG following an *alternating-offers* protocol, alternating between proposal states $\mathcal{S}_p$ and responses states $\mathcal{S}_r$ then: **Assumption 1** *For every $t \leq T$, there exist a probability density function (pdf), $\tau : \mathcal{S}_t \to D(\mathcal{S}_t)$, where $D(\mathcal{S}_t)$ denotes the set of probability distributions over the state space $\mathcal{S}_t$.* **Assumption 2** *The state transition pdf is unknown and Markovian.* **Assumption 3** *For every $t \leq T$, there exist a stochastic reward function $\phi(\mathcal{S}, \mathcal{A})$ : $\mathcal{S} \mathcal{X} \mathcal{A} \to \mathbb{R}^n$.* To link CBG with TBSG, we extend the CBG tuple in Definition 1 and define a new supra-game including stochastic components.

**Definition 2 (Turn-based stochastic game Shapley (1953))** *Is defined by the tuple $(N, T, \mathcal{A}, \delta, \mathcal{S}, R, \tau)$ where $N = \{n_p, n_r\}, (n_p \cap n_r) = \emptyset$ is the set of agents taking turns to propose $n_p$ and respond $n_p$ at each $t$. The solution of a TBSG is a policy set composed of the optimal policies $\pi^* = (\pi_1^*, \dots \pi_n^*)$ such that each agent maximizes its reward conditional on the other agent's policy: $\forall i : \pi_i^* \in argmax_{\pi_i'} \mathbb{E}\left[R_i | \pi_i', \pi_{-i}'\right]$ where $\pi_{-i}' = \pi^* \backslash \pi_i$ and $\pi^*$ is a Nash Equilibrium.*

Since TBSG are a subset of sequential stochastic games, we conclude that we have formalized the relationship shown in Figure 1. We have outlined the formalism that allows us to do the mapping between CBG and MARL.

**CONCLUSIONS** While MARL has been used in the context of CBG, the connection between both is not straightforward. For the first time, we have provided the theoretical underpinnings for a principled use of MARL in CBG by lifting the elements of perfect information games and substituting them for a stochastic transition dynamics and an environment. Some open questions for future research are on the study of the convergence of MARL methods in TBSG.

---

[2] A solved game is a game whose outcome (win, lose or draw) can be correctly predicted from any position, assuming that both players play perfectly.

URM STATEMENT

The authors acknowledge that Lucia Cipolina-Kun meets the URM criteria of the ICLR 2023 Tiny Papers Track.

ACKNOWLEDGEMENTS

This work was supported by the UK Engineering and Physical Sciences Research Council (EPSRC) through a Turing AI Fellowship (EP/V022067/1) on Citizen-Centric AI Systems. (https://ccais.soton.ac.uk/). Lucia Cipolina-Kun is funded by British Telecom.

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

## A  APPENDIX

This section provides further definitions and contextual information relevant to the content of the paper.

### A.1  PRELIMINARIES ON MARL AND STOCHASTIC GAMES

#### A.1.1  LEARNING IN COALITIONAL BARGAINING GAMES

The scope of this paper is to shed light on the theoretical fundamentals supporting the he current use of MARL in CBG in a principled way. As stated in Section 1, MARL is an already adopted method for solving CBG and the literature is vast see Chalkiadakis et al. (2011) however, as shown on Section 2 without specifying a certain number of underlying assumptions, CBG are simply combinatorial optimization games that have been solved. In summary, our aim is not to propose the use of MARL on CBG as it is already used *de facto*, but the ultimate motivation behind this paper is to contribute to the *theory* of learning in CBG games.

**Computational complexity of CBG and the use of MARL.** In the general case of combinatorial optimization games, the application of learning methods such as MARL is not straightforward and deserves to be justified. In particular, since most canonical games are already solved, the value added by learning methods is usually sidelined Fudenberg & Tirole (1991). The formalism behind using MARL to approximate the solution of CBG games has been developed on Section 2 where we outlined the stochastic elements of TBSG as a superset of CBG, substantiating the use of MARL. An additional element of justification for the use of approximation methods in CBG is the *computational complexity* of finding a solution in CBG. In particular, as any partition function games, the computational complexity of CBG is $2^N$. However, without the stochastic components considered, this complexity has already been successfully tackled by works like Rahwan et al. (2009) among others.

#### A.1.2  STOCHASTIC GAMES

A stochastic game (Shapley, 1953) generalises Markov Decision Processes to involve multiple agents, which is why they are the mathematical framework for MARL [3]. Stochastic games are defined as a tuple $<N, S, A, T, R, \gamma>$ where: $N$ denotes the set of n agents, $S$ denotes the set of states, $A = A_i \ldots A_n$ denotes the set of joint actions, where $A_i$ is player $i's$ set of actions. $T : S \times A \to S$ denotes the transition dynamics, $R : S \times A \times N \mapsto R$ denotes the reward function and $\gamma$ denotes the discount factor. The goal is to learn a stationary though possibly stochastic policy, $\pi : \mathbb{S} \mapsto \mathbb{A}$, that maps states to a probability distribution over its actions. We want to find such a policy that maximizes the agent's discounted future reward.

---

[3]MARL is a useful computational framework for stochastic games when the transition dynamics are unknown.

### A.1.3 Connection Between MARL and Stochastic Games

Multi-Agent Reinforcement Learning (MARL) tackles the challenge of learning optimal behaviour by engaging in trial and error interactions within a dynamic multi-agent environment, where the environment dynamics and reward function is unknown. To model the multi-agent environment interaction, MARL adopts the game theoretic model of stochastic games, which are essentially n-agent Markov Decision Processes. In MARL, we are interested in learning a stationary stochastic policy that maps the game's states to a probability distribution over the agent's actions. The goal is to find a policy that maximizes the agent's discounted future reward. The connection between MARL and stochastic games is further discussed in Littman (1994); Bowling & Veloso (2000)

The image below depicts the relationship between the different categories of stochastic games and CBG.

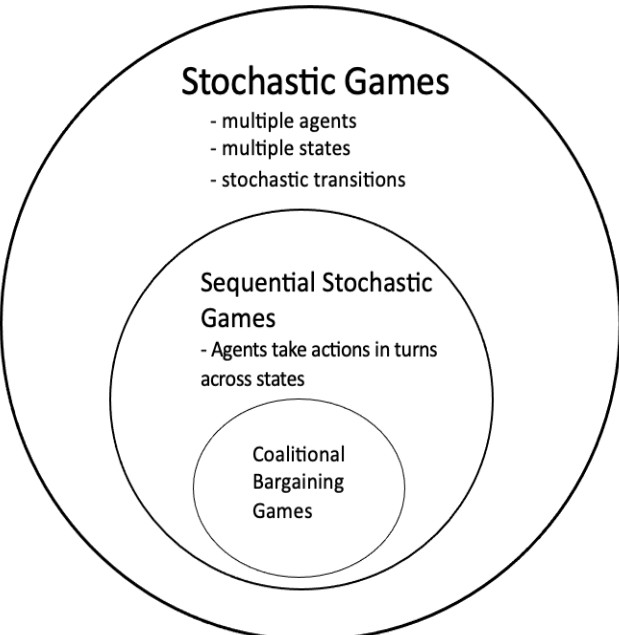

Figure 2: Relationship between stochastic games, sequential stochastic games and coalitional bargaining games - extended.

### A.2 Preliminaries on Coalition Bargaining Games

### A.2.1 The Coalition Structure Generation Problem

**Definition 3** *A coalitional game with transferable utilities is defined by the tuple $(N, v)$ where $N$ is the set of agents and $v(C)$, is a characteristic function $v : 2^C \mapsto \mathbb{R}$ that returns the value that a subset $C \subseteq N$ of agents can obtain acting as a coalition.*

Given a coalitional game $(N, v)$, the Coalition Structure Generation (CSG) problem focuses on generating a coalition structure (as a partitioning on the set of agents $N$) with desirable properties, e.g., those that yield a maximum value.

**Definition 4** *Given a coalitional game $(N, v)$, a coalition structure $\mathbb{C} = \{C_1, \ldots, C_m\}$, is a partition of $N$, with $m \leq n$. That is, for arbitrary distinct $1 \leq k, l \leq m$, we have that $C_k \subset N$, $C_k \cap S_l = \emptyset$, $(k \neq l)$, and $\cup_{k=1}^m C_k = N$.*

As stated in Section A.1.1 finding the optimal coalition structure is of *exponential complexity* in the number of agents. For example, consider the set of agents $N =$

$\{A, B, C\}$, the enumeration of possible *coalition structures* (i.e. set of coalitions) is: $\{\{A\}, \{B\}, \{C\}\}, \{\{A, B\}, \{C\}\}, \{\{A\}, \{B, C\}\}, \{\{A, C\}, \{B\}, \{A, B, C\}\}$.

In what follows, we present the most common bargaining protocols used to solve the coalition structure generation problem.

### A.2.2 Non-cooperative Bargaining Theory of Coalition Generation

The non-cooperative game approach to the problem of cooperation was initiated in the seminal works of Nash (1950; 1953), who presented equilibrium results for a finite-horizon two-person bargaining game known as the Nash Program. The approach aims to explain cooperation as the result of individual players' payoff maximization in an equilibrium of a non-cooperative bargaining game that models pre-play negotiations. Nash stated the seminal results that cooperation should be strategically stable. The approach re-examines a widely held view in economics, called the efficiency principle, that a Pareto-efficient allocation of resources can be attained through voluntary bargaining by rational agents if there is neither private information nor bargaining costs. After Nash, the theory centered its attention into extending the result to infinite-horizon bargaining. The work of Rubinstein (1982) introduces the *alternating offers model* as an equilibrium bargaining protocol for two-person infinite-horizon bargaining. The expansion of this model to n-person bargaining came later with the work of several authors, one example is the protocol proposed by Okada (1996), which presents a sequential bargaining game in which players propose coalitions and feasible payoff allocations until an agreement is reached. Under this protocol, an agreement can be reached in one bargaining round if the proposer is chosen randomly.

**Example of an n-person alternating offers bargaining protocol**.

As an example of a sequential bargaining protocol, we describe the one proposed by Okada (1996). This is the most common protocol implemented in the MARL literature by works such as (Bachrach et al., 2020; Chen et al., 2022; Hughes et al., 2020; Taywade, 2021; Mak et al., 2021) (see Chalkiadakis et al. (2011) for a literature review). In the original formulation by Okada, agents bargain over the members of a coalition and simultaneously over the payoff allocation of the coalition surplus. Our aim is not to propose the application of MARL to this particular protocol but just to use it for illustration purposes of an example of a CBG. Without loss of generality, we will only consider the bargaining over coalition members. While Okada considered this CBG as a perfect-information extensive-form game, we will extend it to a TBSG with full observability.

Following Definition 2, the negotiating process of a CBG involves alternating between proposing states $\mathcal{S}_p$ and responding states $\mathcal{S}_r$ on which agents take proposing actions $\mathcal{A}_p$ and responding actions $\mathcal{A}_r$ respectively. The goal is to find a *coalition structure* $\mathbb{C}$ as in Definition 4. The dynamics of the stochastic game are as follows, let $n_t \leq N$ be the set of "active" players who do not belong to any coalitions on round $t$, then $\{\mathbb{C}_t\} \subset \mathbb{C}$ is the *set* of possible coalitions that $n_t$ players can form, and let $C \subset \{\mathbb{C}_t\}$ be one of these possible coalitions. Each episode starts on a proposing state $\mathcal{S}_p$ and as such, two things happen: the environment selects a *proposer* $i \in n_t$, according to a certain probability distribution $\theta(n_t)$, the proposer then takes an action $\mathcal{A}_p$ choosing a coalition *proposal* of $C$ (where $i \in C$). The game transitions into a responding state $\mathcal{S}_r$ in which all nominated agents such that $n_t \subset C$, take a responding action $\mathcal{A}_r$. If all responders accept the coalition proposal $C$, then it is binding, the environment assigns rewards according to $v(C)$ and another round of bargaining starts with $n_{t+1} = C_t \backslash \mathbb{C}$. If any of the responders reject the proposal, the reward for each responding agent is zero, the episode terminates and another bargaining round starts. The negotiation process ends when every player in N joins some coalition (i.e. a stable coalition structure is formed).

The Figure 3 depicts an example of the stages in a coalitional bargaining game. To simplify, we consider a game with only two agents $N = \{i, j\}$ with policies: $\pi_i, \pi_j$ respectively. In the beginning of an episode, the game enters into a proposing state $\mathcal{S}_p$, where the environment selects an agent acting as the proposer, say $i$, who following $\pi_i$ selects an action $\mathcal{A}_p$. Assume the action taken is to propose the coalition $\{i, j\}$. On the next state $\mathcal{S}_r$, the responding agent $j$ follows $\pi_j$ and selects an action $\mathcal{A}_r$. Assume the action is to respond "accept" the proposed coalition and the game terminates. If the selected action were to "reject", then a new bargaining round would start, where a new proposer is selected and a new proposal is made (following the proposer's policy).

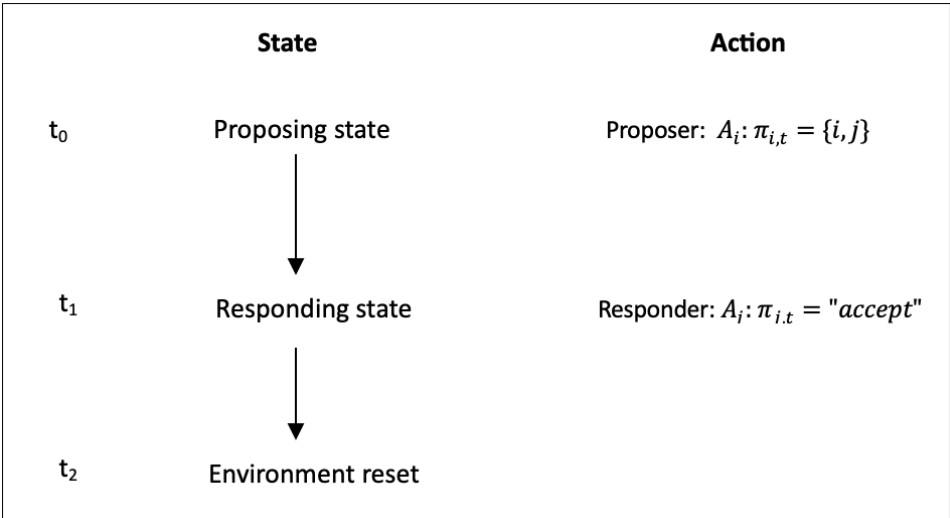

Figure 3: Example of a CBG for $N = \{i, j\}$. An agent is selected at random and makes a coalition proposal following its policy. The other agent in the coalition replies to accept following their own policy. Since all agents have been allocated, the game terminates. Otherwise, the game would continue to a next bargaining round (i.e., a new proposing state).

### A.3 IMPLEMENTATION EXAMPLES

#### A.3.1 LEARNING TO FORM COALITIONS

Following Definition 3, a coalitional game is defined by a tuple $(N, v)$. Consider a game where $N = 2$ and $v$ is as follows:

$$v(\{C\}) = \begin{cases} v(i) = 4.5 \\ v(j) = 4.5 \\ v(i, j) = 16 \end{cases} \tag{1}$$

The reward function states that each agent receives its singleton reward whenever a singleton is formed; while if agents form a coalition, the proceedings $v(i, j)$ are distributed in equal split as follows:

$$R(\{C\}) = \begin{cases} R(i) = 4.5, \text{if } v(i) \\ R(j) = 4.5, \text{if } v(j) \\ R(i, j) = 8 \ \forall i, j, \text{if } v(i, j) \end{cases} \tag{2}$$

Our game setting diverges from that of game theory where agents are assumed to be rational and have full information on the characteristic function of the game. However; in the stochastic game setting, the reward is given by the environment and agents face uncertainty on the game's transition dynamics and the rewards (agents do know know the reward structure in Equation 2, but it is discovered through exploration). We assume full communication among agents. The game proceeds as follows: on each proposing round, an agent is selected at random to be the proposer, its action space is as follows: the agent can choose to remain on its singleton or can propose to form a coalition with the other agent. Following a coalition proposal, the agent in the coalition acts as a responder and can choose to either accept or reject the coalition proposal.

**Expected results.** The characteristic function in Equation 1 defines a *superadditive game* in which, agents gain more through cooperation than acting alone. Thus, it is expected that the grand coalition $v(i, j)$ will form. We expect that in the learned policy, agents learn to propose a coalition to the other agent and the agent learns to accept the coalition proposal.

**MARL results.** Figure 4 shows the mean rewards for each agent trained under 5 different random seeds. Each curve, namely "Policy 0" and "Policy 1" show convergence over a reward corresponding to $R(i, j)$ in Equation 2, where $i = 0$ and $j = 1$. In other words, through bargaining, agents find the most rewarding strategy, without having any information on the preferences or reservation price of other agents.

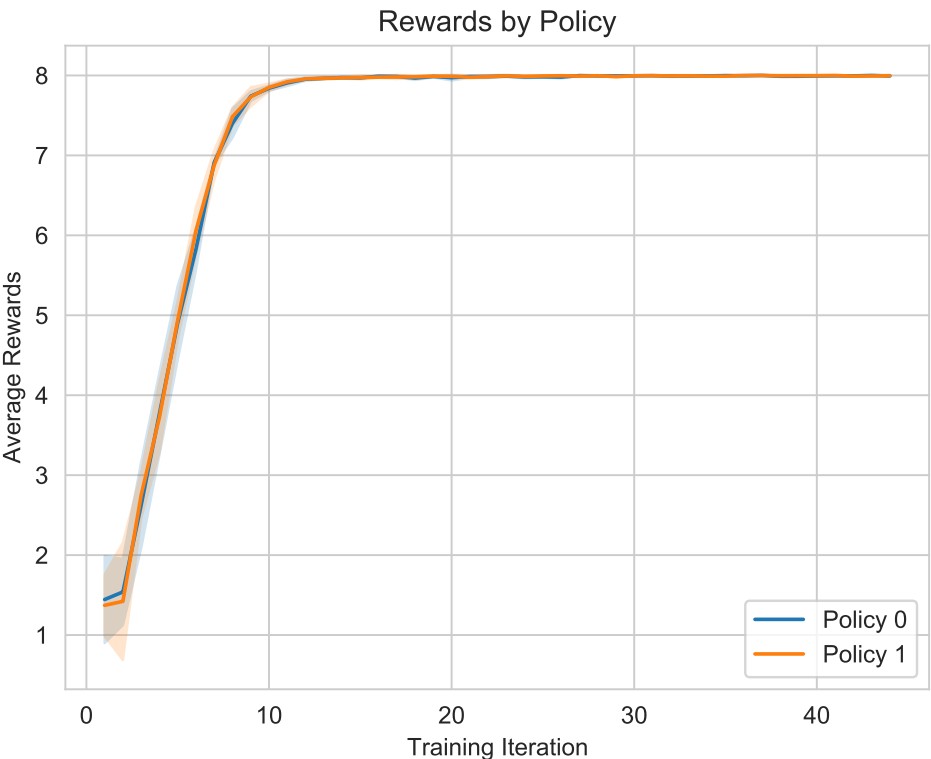

Figure 4: Mean reward obtained by each agent after training their own policies. Agent 0 follows policy 0 and agent 1 follows policy 1. Both agents converge into a reward of cooperation.

### A.3.2   LEARNING TO FORM SINGLETONS

Next, we show a case where agents learn to form singletons. The characteristic function of the game is as follows:

$$v(\{C\}) = \begin{cases} v(i) = 3 \\ v(j) = 11 \\ v(i, j) = 16 \end{cases} \tag{3}$$

The reward function states that each agent receives its singleton reward whenever a singleton is formed; while if agents form a coalition, the proceedings $v(i, j)$ are distributed in equal split as follows:

$$R(\{C\}) = \begin{cases} R(i) = 3, \text{if } v(i) \\ R(j) = 11, \text{if } v(j) \\ R(i, j) = 8 \ \forall i, j, \text{if } v(i, j) \end{cases} \tag{4}$$

**Expected results.** In the characteristic function in Equation 3, even if the game is superadditive, given that rewards are *split equally*, this makes coalition a non-rational action for agent 1. As such, the expected result is that agents learn that cooperation is not the optimal policy.

**MARL results.** Figure 5 shows the mean rewards for each agent trained under 5 different random seeds. Each curve, namely "Policy 0" and "Policy 1" show convergence over a reward corresponding to $R(i), R(j)$ in Equation 4.

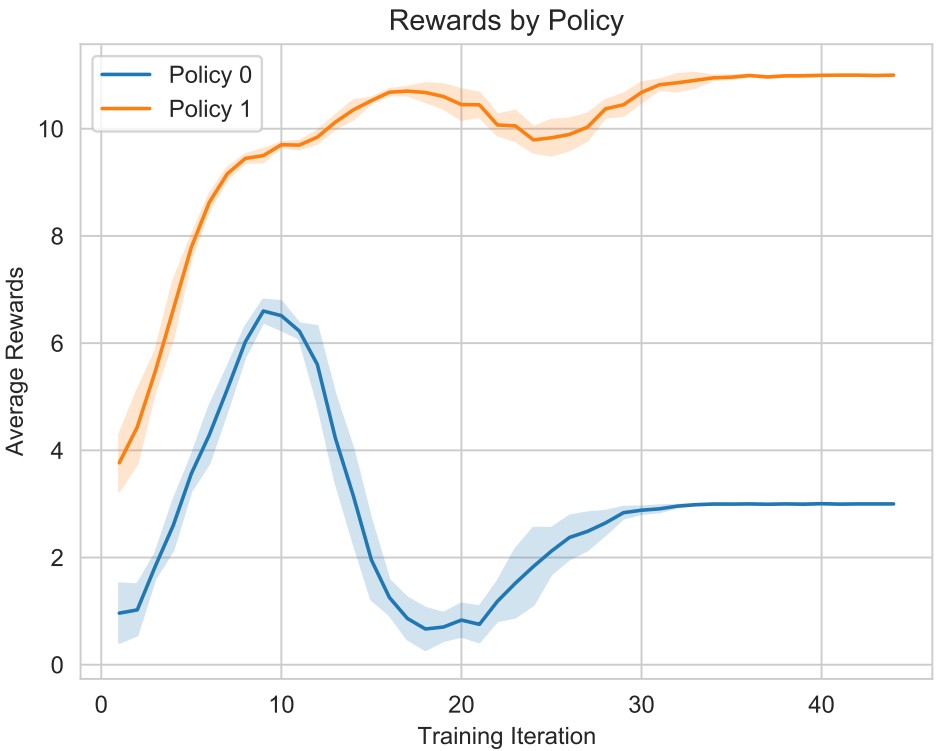

Figure 5: Mean reward obtained by each agent after training their own policies. Agent 0 follows policy 0 and agent 1 follows policy 1. Both agents converge into singleton strategies since cooperation is not beneficial for agent 1.

### A.3.3   LEARNING TO COOPERATE THROUGH BARGAINING OVER REWARDS

The game in Section A.3.2 with the characteristic function in Equation 3 is superadditive, which means that the core is non-empty. However, agents choose to form singletons since the coalition rewards are distributed on the basis of equal split. In this section we show that if agents are free to bargain over the coalition rewards, two emergent behaviours arise: first agents choose to form coalitions over singletons, which is the expected behavior for a superadditive game. Second, the reward is distributed according to the Shapley value.

**Expected results.** If the grand coalition $v(i, j)$ were to form, the Shapley value for each agent is as follows:

$$R(\{C\}) = \begin{cases} R(i) = 4 \\ R(j) = 12 \end{cases} \tag{5}$$

Since the game is superadditive, the core is non-empty and thus, we expect agents to cooperate and distribute the proceedings of $v(i, j)$ according to Equation 5.

**MARL results.** Figure 5 shows the mean rewards for each agent trained under 5 different random seeds. Each curve, namely "Policy 0" and "Policy 1" show convergence over a reward corresponding to $R(i), R(j)$ in Equation 5.

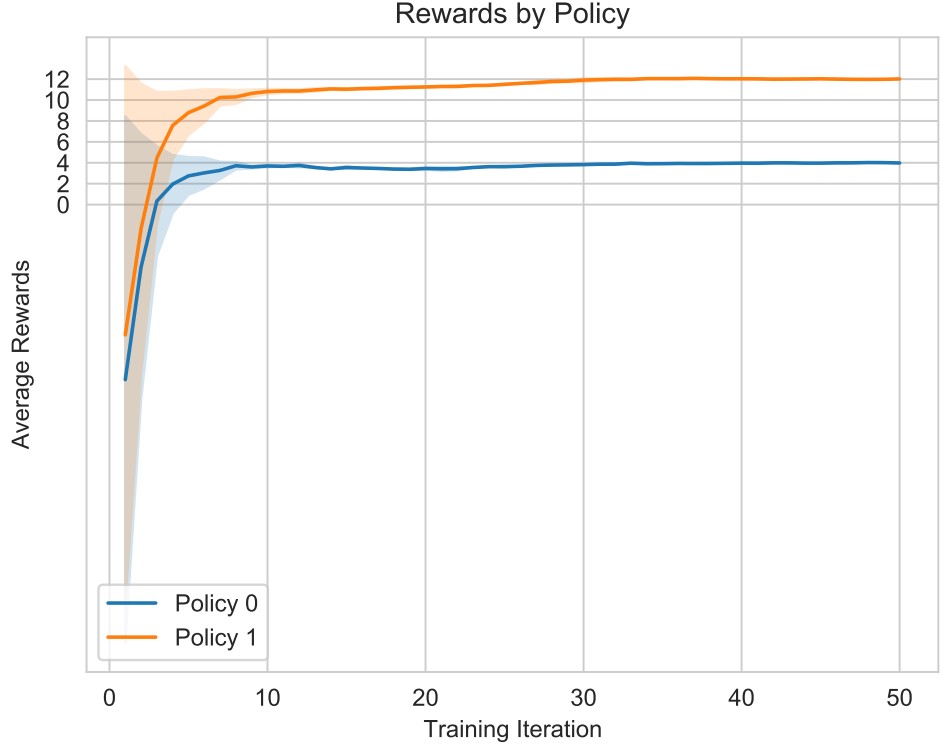

Figure 6: Mean reward obtained by each agent after training their own policies. Agent 0 follows policy 0 and agent 1 follows policy 1. Both agents converge into cooperation strategies and settle for their Shapley value as an emergent behavior.

### A.4 PSEUDOCODE

We present the pseudocode for the application of MARL to a stochastic CBG.

**Result:** Learn to form coalitions
**for** *each episode* **do**
    **while** *unallocated agents exist* **do**
        Select a proposing agent $i$ at random from unallocated agents;
        Agent $i$ proposes a coalition $C$ following its policy;
        **for** *each agent $j$ in proposed coalition $C$* **do**
            Agent $j$ decides to accept or reject following its policy;
            **if** *Agent $j$ accepts* **then**
                Agent $j$ joins the coalition $C$;
            **else**
                Reject the proposal and break;
            **end**
        **end**
        **if** *All agents in proposed coalition $C$ accept* **then**
            Update $Q$ value for all agents in coalition $C$;
        **end**
    **end**
**end**
        **Algorithm 1:** Multi-agent reinforcement learning for coalitional bargaining games

