# OpenReview forum: "MULTI-AGENT REINFORCEMENT LEARNING FOR COALITIONAL BARGAINING GAMES"
_ICLR.cc/2023/TinyPapers — Submitted to Tiny Papers @ ICLR 2023_

### Official Review · Reviewer_Ve1r · 2023-03-19

**Confidence:** 5

**Summary Of Contributions:**

The idea of applying MARL to CBG seems too straightforward and remains questions to me. Beside, there is no experiment to validate the proposed principle.

**Rating:**

Needs Clarification (NC): a submission which does not meet the reviewing criteria and needs clarification for its described problem or solution

**Strengths And Weaknesses:**

Summary:

This paper proposes some theoretical principles for applying MARL to solving coalition bargaining games (CBG). The authors connect MARL to CBG by characterizing a CBG as a turn-based stochastic game (TBSG), which is a subset of stochastic game which can be solved by MARL. Then the authors discuss the benefits and limitations of MARL for solving CBG problems.

Strength:

The main contribution of this paper could be formulating a CBG as a turn-based stochastic game. The paper is easy to follow.

Weakness:

1. Though the authors give the definition of TBSG in Definition 2. But how this TBSG and CBG connect to each other is not clear. For example, in CBG based on the definition in the Appendix, there are generally two status for one agent in the game: proposer and responder. The action sets for a proposer and  a responder are different since proposer need to give a proposal but responder just need to reply accept or reject. But the definition of A does not contain proposals. And the proposing states confuses me. How do you model the state transition between the current state proposal and the next state proposal? Should the state proposal be a choice of an agent, i.e. an action for agents?
2. There is no experiment to validate the proposed principle that MARL can solve a CBG. I think a toy game with a small number of agents would provide the practical evidence for this paper.
3. There are some misunderstanding of MARL. For example, MARL does not guarantee to find an NE for a stochastic game. MARL does not learn optimal policies for each agent given the other agents’ policies. There are some alternative training methods for MARL algorithms, but it’s not all MARL methods doing this way.

**Suggested Changes:**

See my comments in the weakness.

---

### Official Review · Reviewer_EqNR · 2023-03-30

**Confidence:** 3

**Summary Of Contributions:**

The authors aim to show a theoretical link between coalition bargaining games and multi-agent reinforcement to aid in the sound evaluation of reproducibility and generalization of empirical results and formal assessment of the model shortcomings.

**Rating:**

Great Start (GS): a submission which meets some of the reviewing criteria but has room for improvement

**Strengths And Weaknesses:**

The authors have a great idea.
They also have a great start to the abstract and introduction.
The authors briefly explain the domain and gap they intend to address in the literature.

However, they don't have a convincing conclusion of the abstract.  A brief mention of the results and approaches and where they were successful would have made a better conclusion.

The authors overload math terms, and I find the switch between the definition of states and actions confusing. For example, the authors say, "action space from MARL games has been split into two sets of actions (proposals S and actions A)." Earlier, the authors, in definition 2, referred to S as states.

The last paragraph of the appendix and figure (2) should have been in the main body. I think it would have helped shed more light.
In general, even after reading the appendix, I found the paper very high-level, which made it extremely hard to appreciate and fully assess the author's contributions.

Lastly, given the context of the work, experimental results would have improved the paper.

Questions on the sequential responses (not proposals):
- Sequential responses seem suboptimal and expensive. Since the interest is in ensuring all responders agree, wouldn't a broadcast be more efficient?
- I am also curious how the authors would handle responder bias, i.e., responder agreeing because others have.


**Suggested Changes:**

I suggest the authors improve the writeup flow and provide a deeper analysis, including but not limited to consistent and soundproof math, algorithms and experiments to support their contributions.

---

### Comment · Area_Chair_DonE · 2023-06-07
**Final Decision of Paper10 by Area Chair DonE**

After a round of revision, the authors have resolved the concerns of reviewers. Now this work meets the threshold for archival, contents the URM statement and is deanonymized. I would recommend to archive this tiny paper.

---

### Meta-Review · Area_Chair_DonE · 2023-04-07

**Recommendation:** Invite to revise
**Confidence:** 5

**Metareview:**

Overall, the paper has a clear idea but lacks in-depth analysis and experimental validation. The definition of CBG and its connection to MARL need further clarification. The lack of experiments to validate the proposed principles and the confusion regarding MARL's capabilities and limitations are also weaknesses of the paper.

Suggested Changes:

The authors should provide a more detailed explanation of how MARL and CBG connect to each other, including the modeling of states and actions. They should also conduct experiments to validate the proposed principles and provide a practical demonstration of MARL's capabilities for solving CBG problems.

**Summary:**

A paper more like a proposal or an abstract but without any experiments.

**Comments And Feedback To The Authors:**

Thank you for submitting your paper. Your paper presents a novel approach to connecting coalition bargaining games with multi-agent reinforcement learning, and you have made a valuable contribution to the field by characterizing CBG as a subset of TBSG and discussing the benefits and limitations of using MARL for solving CBG problems.

However, I have some feedback and suggestions for improvement. Firstly, the paper lacks in-depth analysis and experimental validation, which would significantly improve the paper's impact and validity. Secondly, some of the definitions and concepts require further clarification to avoid confusion, and I suggest revising these sections. Finally, the paper would benefit from a more detailed conclusion summarizing the results and approaches used in the study.

Overall, your paper has potential, and your ideas are interesting and have significant implications for future research in this area. Thank you again for submitting your work, and I look forward to seeing further developments in this field.

**Reason For Not Giving A Higher Recommendation:**

The paper lacks in-depth analysis and experimental validation, and the definitions of CBG and its connection to MARL need further clarification. Additionally, the lack of experiments to validate the proposed principles and the confusion regarding MARL's capabilities and limitations are also weaknesses of the paper.

**Reason For Not Giving A Lower Recommendation:**

N/A

---

### Decision · Program_Chairs · 2023-04-10

Revision accepted; invite to archive